# Revisiting the contribution of different factors in determining the changes in potential evapotranspiration over China

**Yuchen Shen**[1]☯, **Qirong Wang**[2]☯, **Zhongxiao Feng**[2], **Wei Li**[2]*

**1** Wuxi Meteorological Bureau, Wuxi, Jiangsu, China, **2** Collaborative Innovation Center on Forecast and Evaluation of Meteorological Disaster (CIC-FEMD), Key Laboratory of Meteorological Disaster of Ministry of Education, Joint International Research Laboratory of Climate and Environment Change, Nanjing University of Information Science and Technology, Nanjing, Jiangsu, China

☯ These authors contributed equally to this work.
* weili@nuist.edu.cn

**Data Availability Statement:** The data underlying the results presented in the study are available from https://data.cma.cn/en.

## Abstract

In this paper, a daily gridded observation data across China from 1961 to 2022 were used to calculate daily potential evapotranspiration (PET). The observed variables included daily temperature, sunshine hours, average wind speed, and average relative humidity. PET was determined using the Penman-Monteith method recommended by the Food and Agriculture Organization (FAO). The long-term trend of PET was investigated in six regions of China during different seasons. To further compressed the influence of various meteorological factors on the PET trend, the contribution of each meteorological element to the long-term trend of PET was analyzed. The results indicate the following: (1) PET reaches its peak during summer which values from 145 to 640 mm, while it is lowest during winter from 21 to 244 mm. (2) The spatial patterns of PET trend changes are relatively similar across the four seasons, characterized by a decrease in the eastern regions and an increase in the western regions. The reduction is most significant during the summer and the range of trend is from -2.04 to 1.48 mm/day, while the increase becomes more pronounced in the winter which trend is from -0.34 to 0.53 mm/day. (3) The contribution of factors varies significantly across different regions. In spring and autumn, RH and U have little difference in contribution from other factors. But tsun is varies different from regions, the contribution value is largest in the northwest and smallest in the northeast. However, during summer, tsun become the most significant contributor in the YZ and SE regions, while in winter, $T_m$ emerges as the most significant contributor to the PET trend in all six subregions. In SW, the contribution from $U_2$ is the smallest in all seasons, with RH and $T_m$ being the two crucial factors determining the PET trend in this region.

## 1. Introduction

The interactions and interdependence among the five components of climate system, includes the atmosphere, the hydrosphere, the cryosphere, the lithosphere and the biosphere, are of

**Funding:** This work was funded by the National Key Research and Development Program of China (Grant Nos. 2022YFF0801704) and Youth Foundation of Jiangsu Meteorological Society (Grant KQ202213). The funders had no role in study design, data collection and analysis, decision to publish, or preparation of the manuscript.

**Competing interests:** The authors have declared that no competing interests exist.

great importance. The hydrological cycle serves as a crucial link that binds them together, with evaporation playing a pivotal role. Evaporation is an essential process within the hydrological cycle, contributing to the maintenance of energy and water balance on the Earth's surface [1–3]. In areas with vegetation, evaporation encompasses two processes: the escape of liquid water from the surface of the object, known as surface evaporation, and the transpiration process, where liquid water evaporates within plants and is released into the atmosphere as water vapor through leaf stomata. Potential evapotranspiration (PET) quantifies the amount of water vapor that can occur with ample water vapor supply. It represents the theoretical upper limit of actual evaporation and severs as a metric to characterize the atmosphere's capacity for evaporation [4]. Studying potential evapotranspiration is crucial for assessing climate dryness and wetness [5, 6].

Numerous empirical formulas have been used to calculate PET, with many of them derived from the widely used Penman's formula [7, 8]. The Penman-Monteith (PM) equation, which incorporates the energy balance and mass transport, has been instrumental in determining evapotranspiration in various surfaces such as open water, grassland, and bare soil by using aerodynamic equations [9]. The PM equation accounts for the influence of drag forces on different surfaces, making it a widely utilized equation in PET calculations [10, 11]. In addition to the PM equation, the Thornthwaite method is frequently employed to estimate PET due to its simplicity, as it primarily considers the effect of temperature [12–14]. However, this method is not well-suited for application in the Chinese region since it only accounts for the influence of temperature and disregards radiation, relative humidity, and the wind speed.

In recent years, extensive research has been conducted on the spatial and temporal variation of PET and its underlying causes in China [15–17]. Several studies have contributed to our understanding of these patterns and relationships. Research found that the decrease of PET in China in recent years is primarily attributed to a significant decrease of wind speed, as PET exhibit high sensitivity to this meteorological parameter [18]. Gao et al. investigated ten major basins across the country from 1956 to 2000 and observed a decreasing trend in PET [19]. They found that relative humidity, wind speed, and sunshine hours were closely related to the observed trend of PET. The contribution of evapotranspiration to climate change is particularly important in northern China, where the climate is dominant by the arid condition. Xie and Wang noted that the amount of PET varies across seasons, and the decrease in solar radiation is the main reason for the decrease of PET in northern China in summer [20]. Zhu et al., focused on the Hengduan Mountains from 1960 to 2009 and discovered that increasing temperature led to an increase in PET during autumn and winter [21]. Conversely, decreasing wind speed and sunshine hours were responsible for the decrease in PET during spring and summer. Lang et al., investigated southwest China from 1970 to 2003 and determined that wind speed was the primary factor causing the decrease of PET in spring, autumn and winter in, while the decrease in sunshine hours was the main driver in summer [22]. Other researchers have also examined the sensitivity of PET to various meteorological factors. Liu et al., found that the PET exhibited the highest water vapor pressure, while its sensitivity to minimum temperature was the lowest [23]. Zeng et al., used the principal component analysis to separate the influence of different factors [24]. The first factor comprised thermodynamic factor such as mean temperature, mean water vapor pressure, and latitude. The second factor encompassed radiation-related factors, including humidity, precipitation, and sunshine hours.

Despite numerous studies have examined changes in PET, most of them have focused on specific regions, and there is a lack of comprehensive studies covering the entire country. Additionally, previous research has primarily concentrated on the monthly scale, which fails to capture variation on smaller temporal scales. This study addresses these limitations and contributes to existing literature in two different ways. Firstly, we investigate the trend of PET

across different seasons at a daily scale, encompassing the entirety of China. Previous studies, such as Yin et al., have predominantly focused on the monthly scale, which provides limited insights into the temporal dynamics of PET [18]. By analyzing PET at a finer temporal scale, we can gain a more detailed understanding of its variation patterns. Secondly, this study examines the relative contributions of various meteorological factors to changes in PET at a national scale, utilizing multiple regression coefficients to calculate the relative contribution of different meteorological factors. This approach allows for a comprehensive assessment of the influence of different meteorological factors on PET changes. To our knowledge, this is the first study to investigate these contributions across the entire country, providing valuable insights into the underlying mechanisms driving PET variability. Thirdly, this paper divided China into six subregions and conducted a detailed analysis of the impact of different meteorological factors on the trend of PET. In summary, this study complements previous research by investigating PET trends at a daily scale across China's six subregions and examining the relative contributions of meteorological factors to PET changes. These contributions enhance our understanding of PET dynamics and provide valuable information for the assessment of water resources and climate change impacts.

The overall objective of this study is to investigate the long-term trend of PET and the contribution from different factors in different seasons across the China. The following questions are expected to be solved: (a) Are there significant trend in PET in four seasons across the China? (b) How do various factors influence the changes in PET; (c) What is the relative contribution of different factors to the observed PET changes. This study is structed as follows. Section 2 describes the data the methods used in this study. Section 3 presents the results obtained from the study. Finally, Section 4 concludes the study with a discussion of the findings.

## 2. Data and method

### 2.1 Observation data

This study used a high-resolution daily gridded meteorological observation dataset across China, covering the period from 1961 to 2022 [25]. The data encompassed variables temperature, sunshine hours, wind speed, relative humidity, and air pressure. Given the regional variations in climate characteristics across China due to its complex terrain, the entire country was divided into six subregions to facilitate a more comprehensive analysis of trends in different regions (Fig 1). The analysis was conducted at the individual grid boxes, considering the four distinct seasons: spring (March-May), summer (June-August), autumn (September-November), and winter (December-February).

### 2.2 Method

**2.2.1 Penman-Monteith method.** In this paper, the PM method recommended by the FAO was employed. This method defines PET as a hypothetical evapotranspiration rate of a reference crop canopy. The PM method assumes certain parameters for the reference crop, including a crop height of 0.12m, a fixed crop surface resistance of 70m/s, and an albedo of 0.23. These parameters are chosen to mimic the evapotranspiration characteristics of green grass with an open surface, uniform height, vigorous growth, complete ground cover and adequate suitable moisture. The modified equation representing this approach is as follows.

$$PET = \frac{0.408\Delta(R_n - G) + \gamma \frac{900}{T_{mean}+273} U_2(e_s - e_a)}{\Delta + \gamma(1 + 0.34U_2)} \tag{1}$$

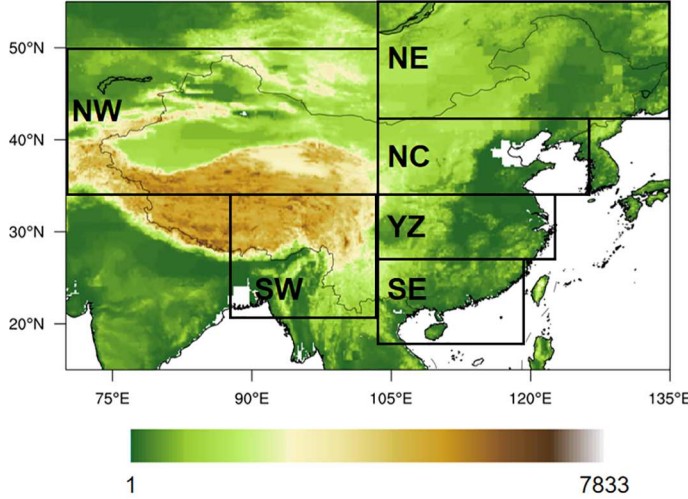

**Fig 1. The six subregions over China in this study: Northwest China (NW: 35˚-55˚N, 74˚-105˚E), Southwest China (SW: 20˚-35˚N, 90˚-105˚E), Northeast China (NE: 42˚-55˚N, 105˚-134˚E), North China (NC: 35˚-42˚N, 105˚-125˚E), Yangtze River basin (YZ: 28˚-35˚N, 105˚-123˚E), and Southeast China (SE: 18˚-28˚N, 105˚-120˚E) (The topographic grid data is download from http://www.ngdc.noaa.gov/mgg/global/relief/ETOPO5/TOPO/ ETOPO5/).** (Made with Natural Earth. Free vector and raster map data @ naturalearthdata.com).

where $\Delta$ is the slope of the saturation water vapor pressure curve ($kPa \cdot {}^{\circ}C^{-1}$), $Rn$ is the net surface radiation ($MJ \cdot m^{-1} \cdot d^{-1}$), $G$ is the soil heat flux ($MJ \cdot m^{-2} \cdot d^{-1}$), $\gamma$ is the psychrometric constant ($kPa \cdot {}^{\circ}C^{-1}$), $Tm$ is the average daily temperature (˚C), $U_2$ is the wind speed at a height of two meters ($m \cdot s^{-1}$). $e_s$ is the saturated water vapor pressure ($kPa$), $e_a$ is the actual water vapor pressure ($kPa$). A detailed introduction to the calculation of PET step by step can refer to Allen et al. [9].

**2.2.2 M-K trend test method.** There are many ways to estimate the linear trend and test the statistical significance, such as estimating the trend using the ordinary least square (OLS) and student's test to test the corresponding significance. This method assumes that the regression residuals are independent Gaussian random variables. A non-parametric method is widely used [23, 24], which includes using Sen's slope to estimate the linear trend, and Mann-Kendall (MK) test to assess the corresponding statistical significance for temporal series. This method has an advantage not requiring any distribution form. The statistic value $S$ is calculated as:

$$S = \sum_{i=1}^{n-1} \sum_{j=i+1}^{n} sgn(x_j - x_i) \tag{2}$$

Where n is the length of record, $x_i$ and $x_j$ are the value in the time series $i$ and $j$ ($j > i$), respectively. The $sgn$ is the sign function calculated as:

$$sgn(x_j - x_i) = \begin{cases} 1, (x_j - x_i) > 0 \\ 0, (x_j - x_i) = 0 \\ -1, (x_j - x_i) < 0 \end{cases}, i = 1, 2, \ldots j \tag{3}$$

The S value is transferred to standard normal form using Eq (6):

$$Z_s = \begin{cases} \dfrac{S-1}{\sqrt{Var(S)}}, & S > 0 \\ 0, & S = 0 \\ \dfrac{S+1}{\sqrt{Var(S)}}, & S < 0 \end{cases} \qquad (4)$$

Where $Var(S)$ denotes the variance of S, the positive value of $Z_s$ indicate the increasing trend for PET while negative $Z_s$ denotes the decreasing trend. The null hypothesis with no monotonic trend exists in time series is rejected and a significant trend is declared if $|Z_s| > Z_{(1-\alpha)}$. We used $\alpha = 0.05$ in this study, the null hypothesis is rejected at 5% significance level if $|Z_s| > 1.96$. We applied MK test on PET temporal series at individual sites in spring, summer, autumn and winter from 1961 to 2022.

**2.2.3 Relative contribution analysis.** Multiple Linear Regression (MLR) is a statistical method used to model the relationship between a dependent variable and two or more independent variables. Unlike simple linear regression, which considers the relationship between two variables, MLR extends the analysis to multiple predictors, allowing for a more comprehensive understanding of the factors influencing the dependent variable.

In a multiple linear regression model, the relationship between the dependent variable (Y) and the independent variables (, $X_2$, . . ., $X_n$) is represented by the equation:

$$Y = a_0 + a_1 X_1 + a_2 X_1 + \ldots + a_n X_n + \varepsilon \qquad (5)$$

Where $Y$ is is the dependent variable, $X_1, X_2, \ldots, X_n$ are the independent variables. $a_0$ is the intercept (the value of $Y$ when all $X$ variables are zero). $a_1, a_2, \ldots, a_n$ are the coefficients representing the change in $Y$ for a one-unit change in the corresponding $X$ variable. $\varepsilon$ is the error term, representing the unexplained variability in $Y$.

The goal of multiple linear regression is to estimate the coefficients ($a_1, a_2, \ldots, a_n$) that best fit the data, minimizing the sum of squared differences between the predicted and observed values. In our research, it was used to investigate the contribution of five factors, namely temperature ($T_m$), wind speed ($U_2$), sunshine duration (tsun) and relative humidity (RH), to the PET changes in different seasons. This method is widely used in previous studies [26].

$$PET = a_1 T_m + a_2 tsun + a_3 U_2 + a_4 RH + C \qquad (6)$$

Where $a_1, \ldots a_4$ are regression coefficients, and $C$ is a constant. The contribution of different factors to the PET changes can be achieved:

$$C_k = \frac{a_k \Delta F_k}{\Delta PET} \qquad (7)$$

$a_k$ represents the $kth$ regression coefficient, and $\Delta F_k$ is changes of the $kth$ factor between the last 10 years and first 10 years, $\Delta PET$ is changes of PET between the last 10 years and first 10 years. The relative contribution can be calculated based on Eq (7), for example: the relative

contribution of $T_m$ can be calculated as Eqs (8) and (9) shows

$$R_k = \frac{|C_k|}{\sum_{i=1}^{k} |C_k|} \times 100 \tag{8}$$

$$C_{T_m} = \frac{a_1 \left( T_{m(2013-2022)} - T_{m(1961-1970)} \right)}{PET_{(2013-2022)} - PET_{(1961-1970)}} \tag{9}$$

$$R_{T_{max}} = \frac{|C_{T_m}|}{\sum_{i=1}^{k} |C_k|} \times 100 \tag{10}$$

## 3. Results

### 3.1 Climatology of PET

Different regions in China have different climate characteristics. Understanding the distribution of PET climatology in these different regions can help us understand its changes in the region better. Fig 2 shows the climatology pattern of PET averaged from 1991–2010 in four seasons. Notably, the pattern of annual PET is consistent with that of spring and summer, therefore the annual PET is excluded from our study. In spring, the patterns of PET in spring shows distinct characteristics, with higher values observed in northern China with highest value reaching 428 mm and lower values in southern China with smallest value reaching144 mm. The regions with highest PET values are concentrated in northwest and north China, particularly in northern Xinjiang and northwestern Inner Mongolia. Conversely, the regions with lower PET values are primarily located in the Yangtze River basin and southeast China. However, in southwest China, particularly in Yunnan Province, the PET values surpass those of other regions at the same latitude, some gridded boxes in this area record PET value exceeding 300mm. This indicates that southwest China experiences relatively higher evapotranspiration rates compared to other regions with similar latitudes.

Generally, the PET in summer is the highest among the four seasons, spanning a range of 120–640 mm across the entire country. The regions with the maximum PET values are predominantly found in northern Xinjiang and northern Inner Mongolia. In many parts of country, PET exceeds 200 mm, except for southwest China. Because of the combination effect of longer sunshine hours, higher average temperature and lower relative humidity than other regions, Northwestern China received the highest PET values, with majority of regions reporting PET values exceeding 360mm. Unlike in spring, the PET values in southwest China are comparatively lower than those in regions at the same latitude, this may due to higher relative humidity than other regions. In Northern China and Northeastern China, PET values generally exceed 240mm.

In autumn, there is a relatively small spatial variation in PET across China, with values ranging from 80 to 320mm. Some regions in northwestern China and the southeast coast of China report PET values exceeding 300 mm. Notably, the PET in the southeastern coastal region is higher compared to areas at the same latitude, this is may partly due to the influence of ocean at low latitude. Conversely, southwestern China and north of northern China receive the lowest PET values.

The PET in winter is the smallest among the four seasons, with the majority of grids across the country recording PET values of less than 140mm. The overall spatial distribution of PET in winter is opposite to that of spring, with the pattern characterized by higher values in south

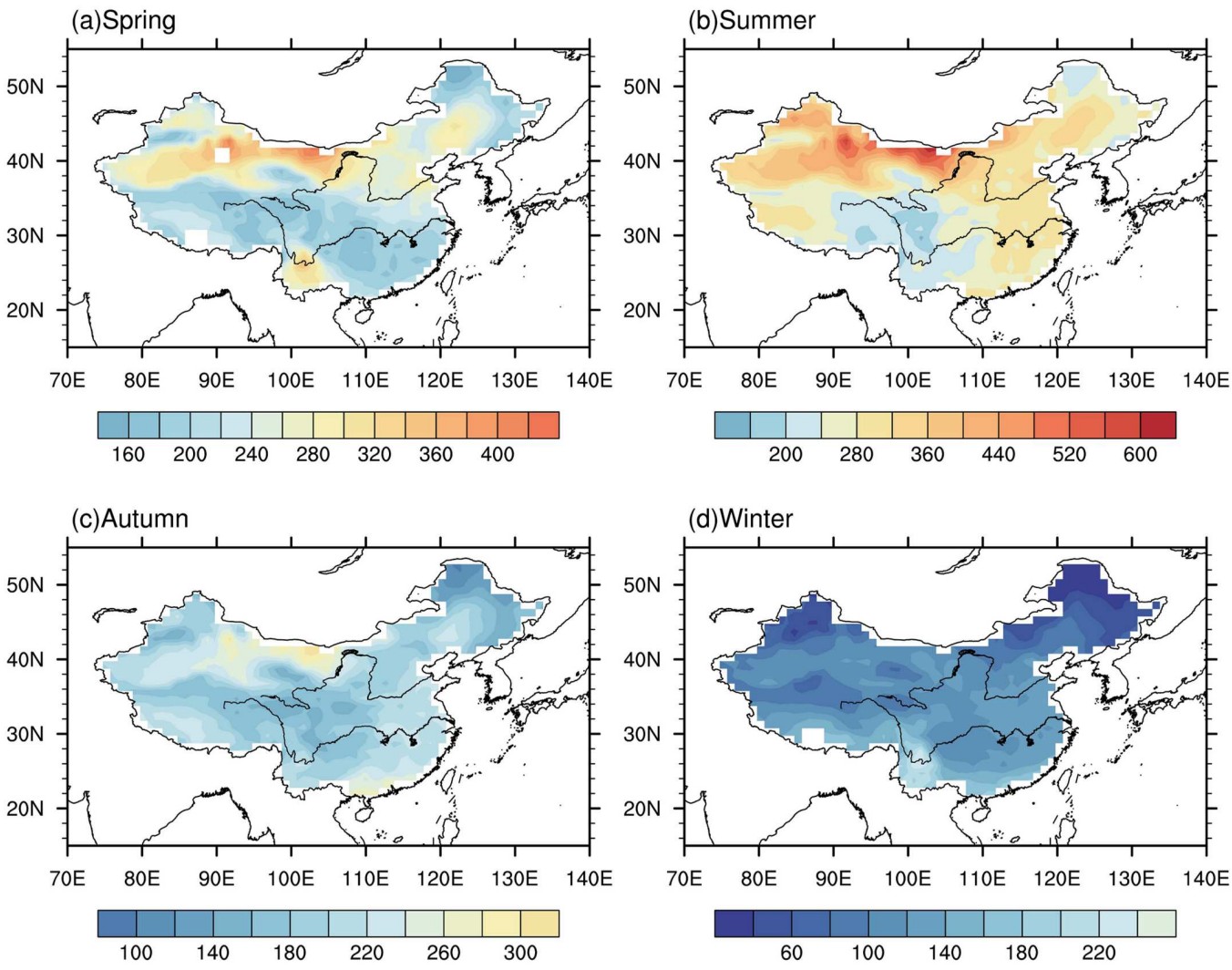

**Fig 2. Spatial distribution of climatology of PET (mm) in spring (A), summer (B), autumn (C) and winter (D) averaged from 1986 to 2005.** (Made with Natural Earth. Free vector and raster map data @ naturalearthdata.com).

China and lower value in north China. In contrast to other seasons, where the areas with the maximum PET are primarily located in northern China, the regions receiving the highest PET values in winter are found in South China.

In summary, PET exhibits a seasonal variation, with the largest value reaching 630 mm occurring during summer and the smallest value with 21 mm occurring during winter. Northwestern China generally experiences higher PET compared to other regions, except in winter. Of particular interest is the PET in southwest China, specifically in Yunnan Province, which demonstrates a distinct pattern of "large, small, small and large" across the four seasons of spring, summer, autumn and winter, respectively, in comparison to areas at the same latitude.

### 3.2 Trend of PET

We conducted M-K test, as described in section 2.2.2, to assess the statistically significant of the PET trend in spring, summer, autumn and winter from 1961 to 2022. Fig 3 display the

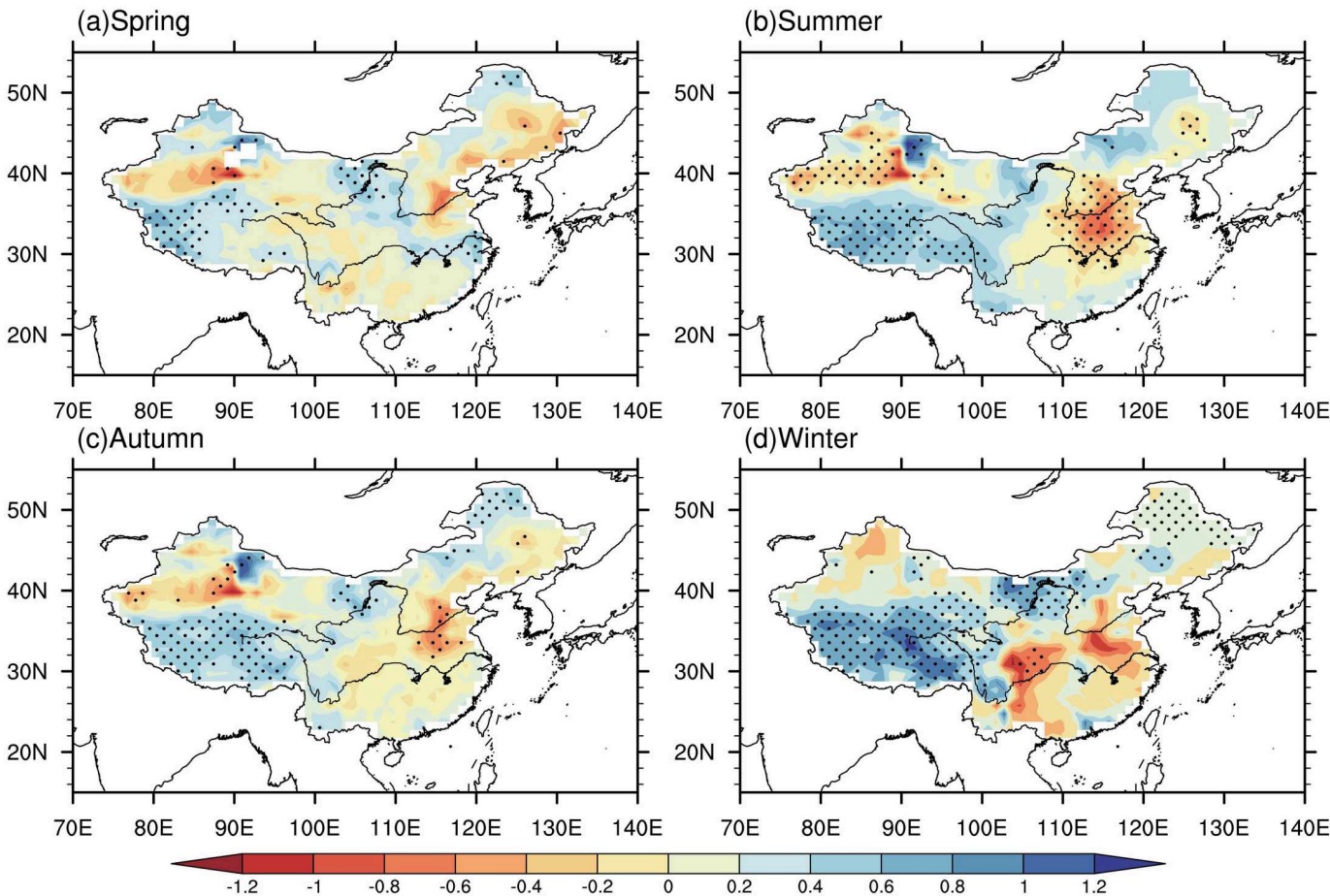

**Fig 3. Spatial distribution of trend of PET (mm / year) in spring (A), summer (B), autumn (C), and winter (D) from 1961 to 2022, with blue (red) indicating an increasing (decreasing) trend.** Solid cycles indicate that the trend passed the test at a significance level of 0.05. (Made with Natural Earth. Free vector and raster map data @ naturalearthdata.com).

trend pattern of PET in four seasons. In general, the trends remain relatively consistent across the four seasons. The eastern regions predominantly witness a decreasing trend, whereas the western regions predominantly show an increasing trend. In spring, the areas showing significant increasing trend is higher than those showing a significant decreasing trend. The areas with significant decreasing trend are mainly located in parts of Northern China and northeastern China, while the areas over Yangtze River basin, western of Tibetan and eastern Inner Mongolia exhibit a significant increasing trend.

Compared to spring, the summer shows a significantly wider range of notable trends. The entire Tibetan region exhibits a significant increase trend. In much of northern China and the northern parts of Xinjiang, there are significant decreasing trends. The areas showing significant decreasing trends in autumn has narrowed compared to summer, and their magnitude has also decreased, particularly in the North China region. The increase in PET in the winter season in the Tibetan region and eastern Inner Mongolia exhibit the strongest among the four seasons, in term of significant and magnitude of trend. The magnitude of trend can reach to more than 12 mm/decadal. The significant decreasing trend in the North China region has disappeared, but there is a significant decrease in PET in the Sichuan Basin.

Overall, the trend patterns are consistent between four seasons with decreasing in eastern China and increasing in the western China. The areas showing significant increasing trend are concentrated in Tibetan region, and the areas exhibiting decreasing trend are concentrated in the North China and western Xinjiang. It is consistent with the warming and humidification trends observed in northwestern China during this period. Conversely, southwestern China has experienced a significant increasing trend in PET in all seasons, indicating a higher likelihood of drought events occurring in this region [27, 28]. These findings align with previous studies highlighting the potential impacts of climate change on the water availability and drought vulnerability of southwestern China.

## 3.3 Trend of different meteorological factors

The spatial pattern and magnitude of PET trend are closely linked to changes in meteorological factors that influence PET. The absence of significant trend changes in certain areas may be attributed to the counterbalancing effects of different meteorological factors. In this section, we examine the trends of four meteorological factors-temperature, sunshine hours, wind speed and relative humidity-in six subregions of China over a 62-year period in four seasons (Fig 4). The histograms illustrate the magnitude of trends for each meteorological factor in different regions, with positive values indicating an increasing trend and negative values indicating a decreasing trend.

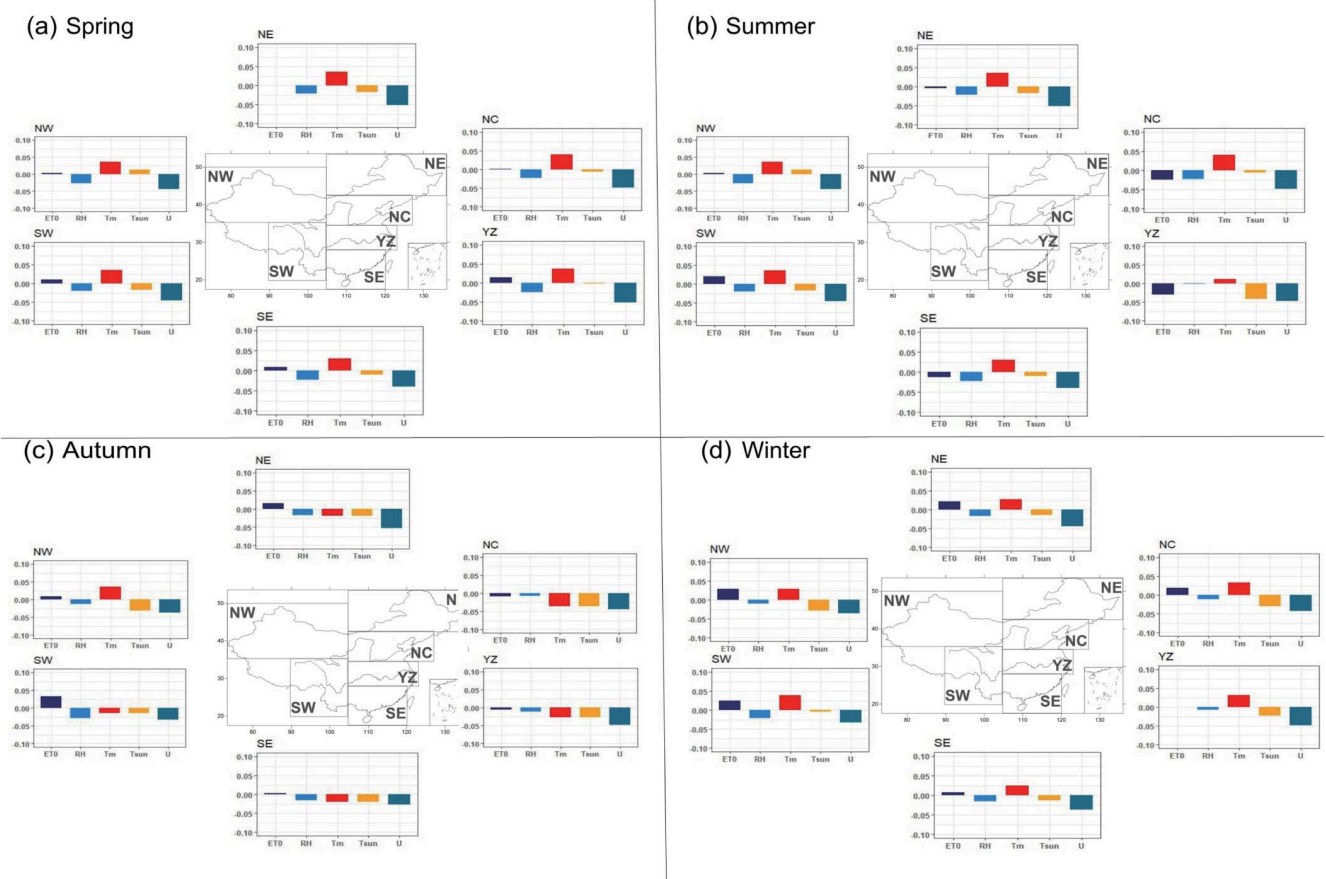

**Fig 4. The averaged trends of PET (ET0), relative humidity (RH), mean temperature ($T_m$), sunshine hours (tsun) and wind speed ($U_2$) for spring (A), summer (B), autumn (C) and winter (D) from 1961 to 2022.** (Made with Natural Earth. Free vector and raster map data @ naturalearthdata.com).

From Fig 4, it is evident that $U_2$ and RH exhibit a decreasing trend in all six subregions during spring (Fig 4A), which is consistent with the decreasing trend of PET. Among the factors, $U_2$ received the most substantial magnitude of decreasing trend in all subregions, followed by the RH. Specifically, the decreasing trend in tsun is most pronounced in the NE, followed by SW and SE. While the displaying an increasing trend in NW. The trend of RH indicates a similar magnitude among six subregions. On the other hand, $T_m$ exhibit increasing trend throughout the entire regions.

Fig 4B demonstrates similar trend patterns in summer. Both $U_2$ and RH show a decreasing trend, while $T_m$ show a slightly increasing trend in all six subregions. The trend of tsun exhibits decreasing trend, except for the NW where an increasing trend is observed. Among the meteorological factors, $U_2$ continues to show the largest decreasing trend, followed by the tsun. The YZ witnesses the largest decreasing trend in tsun, while the weakest trend is found for RH.

In the autumn (Fig 4C), the decreasing trend of $U_2$ remains the most significant factor determining the decreasing of PET, especially over the NE. The smallest trend of $U_2$ is observed in the SW, where the RH shows the largest decreasing trend. This suggests that the changes in RH play a dominant role in the decrease of PET in SW. tsun exhibit decreasing trend in all six subregions with larger magnitude in northern China than in southern China. The tsun exhibits the largest decrease in the NC, followed by the NW, while the smallest trend is found in the SW region.

In winter (Fig 4D), similar to other three seasons, the trend of $U_2$ is the largest among the five meteorological factors, with a pattern of higher magnitude in the east and lower in the west. The RH exhibit no trend in NW and SW. A relatively large trend of tsun is observed in NC, YZ and SW, while a relatively small trend of tsun is found in SE and NE.

Comparing the changes in different seasons, the trend of $U_2$ in spring is generally greater than other seasons in all six subregions. The trend of $T_{min}$ in SW is the greatest among six subregions in four seasons. In summer, the decreasing trend of tsun is larger in NC, YZ and SE compared to spring. Additionally, the magnitude of the trend of $T_{max}$ in cold season (autumn and winter) is greater than in warm season (spring and summer).

## 3.4 Contribution from different meteorological factors to PET trend

In order to quantify the relative contribution of different meteorological factors to the trend of PET in four seasons, we used the method described in section 2.2.3 to calculate the relative contribution from $T_m$, $U_2$, tsun and RH to the trend of PET (Fig 5).

The decreasing in PET during spring (Fig 3A) is primarily attributed to the deceasing of $U_2$ and increasing of $T_m$ (Fig 4A), which accounts for approximately 20% (SE) to 40% (NC) of the relative contribution across six subregions(Fig 5A). These findings align with a study by Yin et al. (2010) which identified wind speed as the primary variable causing the decreasing trend of PET. The decreasing of RH also plays an important role in changes of PET, the contribution from RH is comparable to the contribution from $U_2$ in YZ and SE, the opposite effect of those two factors ultimately lead to weak trend of PET (Fig 4A). This also demonstrated that the effect of RH on the PET in humid region is quite large and should be considered. The contribution of $T_m$ is greater than RH in NE and NW, while the contribution from RH is larger than $T_m$ in the rest of the regions (Fig 5A). The contribution of tsun is relatively low, with the smallest contribution observed in NC and NW (5%) and the largest contribution in SE (30%). For the specific region, tsun and RH played a crucial role in the changes of PET in SE. while RH is the most contributing factor in the YZ. It can be seen that the decrease of PET is primarily driven by the decrease of $U_2$ and increasing of $T_m$, while the changes in PET in Northern

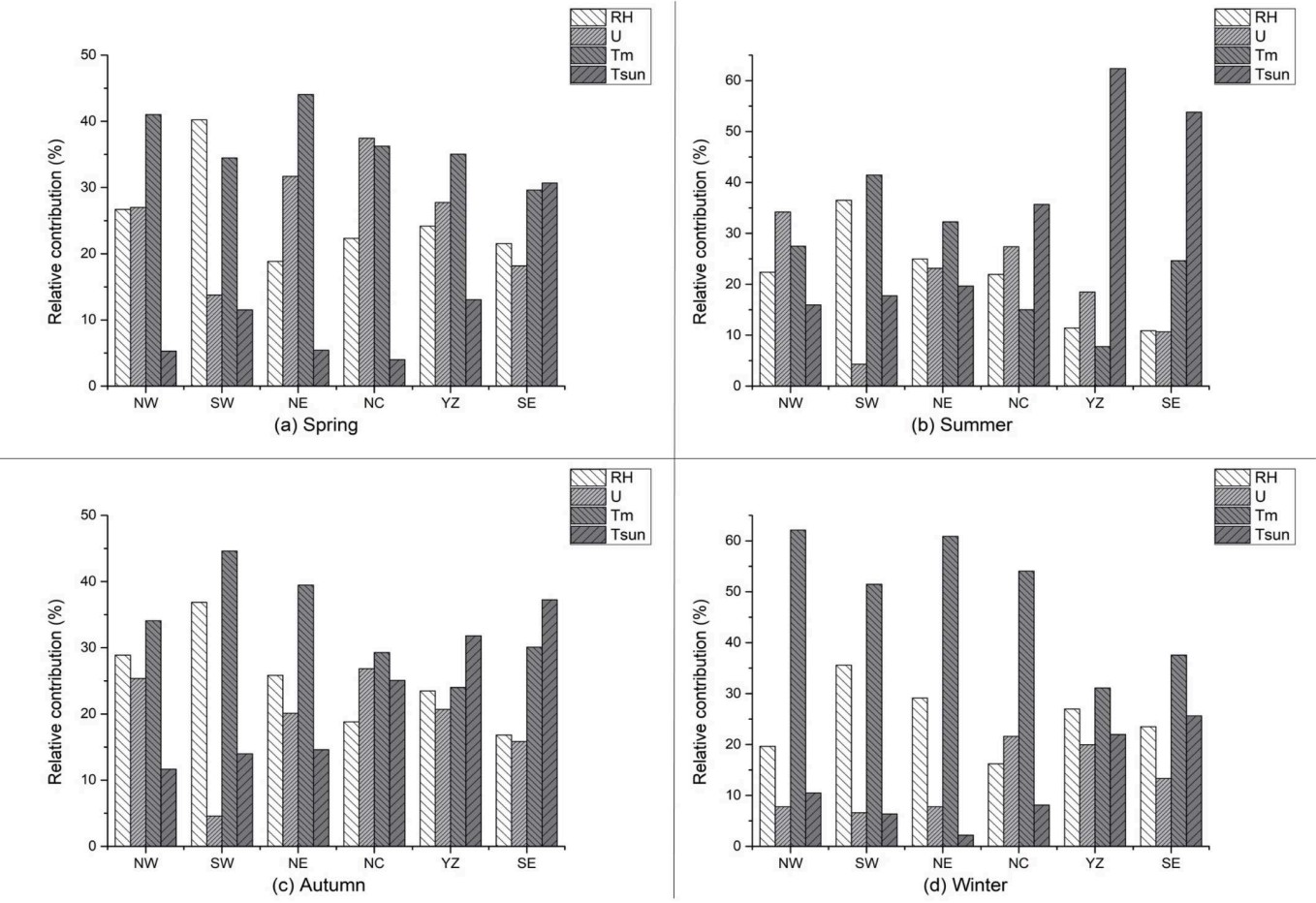

**Fig 5. Relative contribution (%) of different factors to changes in PET in six subregions of China in spring (A), summer (B), autumn (C), and winter (D) from 1962 to 2022.**

China remain relatively stable due to the combined effect of the decreasing of wind speed and decreasing of RH.

In summer, the contribution of factors varies across different regions. Such as, $U_2$ is an important contributing factor for the NW region, where their contribution surpasses that of other factors, reaching up to more than 35% (Fig 5B). The $T_m$ is the largest contributors in SW and NE. Additionally, in YZ and SE, the contribution of tsun is greater than other regions, with values exceeding 20%. Consequently, the little changes of PET in NE and NW are primarily attributed to offset effects among different factors. In NC and YZ, the combined effects of $U_2$ and tsun contribute to the decrease in PET. In YZ, the large reduction in PET is primarily influenced by the decreasing in tsun. In SE, PET remains relatively stable due to the combined effect of tsun, RH and $U_2$ (Fig 4B).

In autumn, tsun continues to be a key contributor to PET changes in humid regions, such as YZ and SE, although its relative contribution has decreased compared to the previous two seasons. The relative contribution of $T_m$ dominate the changes in NW, SW, NE and NC, ranging from 30% in NC to 45% in SW across six subregions. $U_2$ remain the smallest factors driving PET changes in SW (nearly 5%). The largest contribution from $U_2$ is found in NC and NW, reaching 25%. The contribution from RH to the PET changes exhibit a little difference among

various regions, with the largest contribution over SW reaching to nearly 40%. The notable increasing trend in PET in SW (Fig 4C) is primarily attributed to the combined effect of increasing in $T_m$ and decreasing in RH. Although tsun also makes a significant contribution in YZ and SE, factors with an equal and opposite contribution offset their impact on PET reduction, resulting in minimal PET changes.

PET in NE remains relatively stable due to the combined contribution from the decrease in wind speed and the increase in $T_m$. The increasing trend of PET in SW and SE is mainly driven by the substantial contribution from the increase in $T_m$ and decrease in RH.

In winter, $T_m$ emerges as the most significant contributor to the PET trend in all six subregions, with relative contributions ranging from 40% in SE to 70% in NE across six subregions (Fig 5D). This highlights the substantial influence of temperature factors on the PET during winter. On the other hand, tsun plays the smallest role among the meteorological factors, with relative contribution below 10% in all subregions, except humid region (YZ and SE). The contribution from tsun becomes the smallest in NW, SW, NE and NC. Although, the contribution from RH is relatively small, it still exhibits greater significance in humid region compared to dry region. As a result, the PET trends in winter displays a weaker pattern in all six subregions due to the intricate interaction among each meteorological factor.

In conclusion, the contribution of factors varies notably across different regions. The contribution from different factors exhibits little difference in spring and autumn. While in summer, the tsun become the largest contribution in YZ and SE. and $T_m$ emerges as the most significant contributor to the PET trend in all six subregions in Winter. The contribution from $U_2$ is smallest in all seasons in SW. RH and $T_m$ are the two important factors determining the trend of PET in this region.

## 4. Conclusion and discussion

In this paper, we calculated daily PET using Penman-Monteith method, recommended by the FAO, based on a daily gridded observation dataset. The analysis covered four seasons (spring, summer, autumn, and winter) and a time period from 1961 to 2022. The trend of PET was investigated in six subregions of China. To better understand the driver of PET trends, the relative contributions of five meteorological factors ($T_m$, $U$ tsun and RH) to the PET trend were determined using the multiple regression method, and the main results are as follows.

1. The climatology of PET shows that it is highest in summer, followed by spring and autumn, and lowest in winter. The spatial pattern of PET trends indicates that there is a higher magnitude of PET in northern China compared to southern China. Specifically, the largest PET values are observed in the northwest region during spring, summer and autumn. However, in winter, the pattern is reversed, and the highest PET values are found in the southeast region.

2. The spatial patterns of PET trend changes are relatively similar across the four seasons, characterized by a decrease in the eastern regions and an increase in the western regions. The reduction is most significant during the summer, while the increase becomes more pronounced in the winter.

3. Among several factors affecting PET, RH exhibits a decreasing trend in almost all regions and seasons, with larger trend changes during the warm year compared to the cold year. Tsun shows the most pronounced decreasing trend during the autumn and winter seasons, especially in the eastern regions. Meanwhile, $U_2$ exhibits a significant decreasing trend in all regions and seasons.

4. The contribution of factors varies significantly across different regions. There is little difference in the contribution from other factors during spring and autumn. However, during summer, tsun become the most significant contributor in the YZ and SE regions, while in winter, $T_m$ emerges as the most significant contributor to the PET trend in all six subregions. In SW, the contribution from $U_2$ is the smallest in all seasons, with RH and $T_m$ being the two crucial factors determining the PET trend in this region.

This paper provides valuable insights into the trends of PET and the contribution of different meteorological factors to these trends. However, there are several areas that could be improved in future research. Firstly, this study has identified the contribution of $T_m$, $U_2$, tsun, RH to the trend of PET, understanding the relative sensitivity of PET to each of these factors is important. It would be beneficial to study the sensitivity of PET to different meteorological factors in more detail, which can be conducted by using sensitivity express [29]. Secondly, the characteristics of drought at the daily scale, such as persistence, intensity, and frequency may be further studied using daily PET. Understanding the interaction between PET, precipitation, and drought dynamics would provide valuable insights into the water balance and hydrological processes [30]. Thirdly, evaluating the performance of CMIP6 models in simulating daily PET is essential for projecting PET under continued global warming [31, 32].

## Acknowledgments

We would like to thank the editor and the anonymous reviewers for their helpful suggestions and comments. We thank the National Meteorological Information Center (NMIC) of China Meteorological Administration for providing us with data. We also thank National Oceanic and Atmosphere Administration (NOAA) for providing us with topographic grid data.

## Author Contributions

**Conceptualization:** Zhongxiao Feng, Wei Li.

**Data curation:** Yuchen Shen, Qirong Wang.

**Formal analysis:** Qirong Wang.

**Funding acquisition:** Wei Li.

**Methodology:** Yuchen Shen, Qirong Wang, Zhongxiao Feng, Wei Li.

**Software:** Qirong Wang, Zhongxiao Feng.

**Supervision:** Wei Li.

**Visualization:** Zhongxiao Feng.

**Writing – original draft:** Yuchen Shen.

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
