## [Decision Letter · Decision Letter 0]

21 Nov 2023

PONE-D-23-30478Revisiting the contribution of different factors in determining the changes in Potential evapotranspiration over ChinaPLOS ONE

Dear Dr. Li,

Thank you for submitting your manuscript to PLOS ONE. After careful consideration, we feel that it has merit but does not fully meet PLOS ONE’s publication criteria as it currently stands. Therefore, we invite you to submit a revised version of the manuscript that addresses the points raised during the review process.

We look forward to receiving your revised manuscript.

Kind regards,

Mohammed Magdy Hamed

Academic Editor

PLOS ONE

Journal Requirements:

"WL

2022YFF0801704

National Key Research and Development Program of China

https://en.most.gov.cn/programmes1/200610/t20061009_36224.htm

NO"

"This work is supported by the National Key Research and Development Program of China (Grant Nos. 2022YFF0801704) and Youth Foundation of Jiangsu Meteorological Society (Grant KQ202213). "

"WL

2022YFF0801704

National Key Research and Development Program of China

https://en.most.gov.cn/programmes1/200610/t20061009_36224.htm

NO"

7. PLOS requires an ORCID iD for the corresponding author in Editorial Manager on papers submitted after December 6th, 2016. Please ensure that you have an ORCID iD and that it is validated in Editorial Manager. To do this, go to ‘Update my Information’ (in the upper left-hand corner of the main menu), and click on the Fetch/Validate link next to the ORCID field. This will take you to the ORCID site and allow you to create a new iD or authenticate a pre-existing iD in Editorial Manager. Please see the following video for instructions on linking an ORCID iD to your Editorial Manager account: https://www.youtube.com/watch?v=_xcclfuvtxQ

8. We note that [Figures 1-3] in your submission contain [map/satellite] images which may be copyrighted. All PLOS content is published under the Creative Commons Attribution License (CC BY 4.0), which means that the manuscript, images, and Supporting Information files will be freely available online, and any third party is permitted to access, download, copy, distribute, and use these materials in any way, even commercially, with proper attribution. For these reasons, we cannot publish previously copyrighted maps or satellite images created using proprietary data, such as Google software (Google Maps, Street View, and Earth). For more information, see our copyright guidelines: http://journals.plos.org/plosone/s/licenses-and-copyright.

a. You may seek permission from the original copyright holder of Figures 1-3 to publish the content specifically under the CC BY 4.0 license.  

9. Please remove your figures from within your manuscript file, leaving only the individual TIFF/EPS image files, uploaded separately. These will be automatically included in the reviewers’ PDF.

Reviewers' comments:

Reviewer's Responses to Questions

**Comments to the Author**

1. Is the manuscript technically sound, and do the data support the conclusions?

Reviewer #1: Yes

Reviewer #2: Yes

2. Has the statistical analysis been performed appropriately and rigorously? 

Reviewer #1: Yes

Reviewer #2: Yes

3. Have the authors made all data underlying the findings in their manuscript fully available?

Reviewer #1: Yes

Reviewer #2: Yes

4. Is the manuscript presented in an intelligible fashion and written in standard English?

Reviewer #1: Yes

Reviewer #2: No

5. Review Comments to the Author

Reviewer #1: Authors are kindly urged to carefully review and address the constructive feedback provided in the manuscript review process. It is imperative that the authors thoroughly consider the points for correction and make the necessary amendments to enhance the quality and clarity of their work. they diligence in addressing these suggestions will significantly contribute to the overall excellence of the paper and its suitability for publication.

Reviewer #2: Review Comments:

This research study was conducted on the topic “Revisiting the contribution of different factors in determining the changes in Potential evapotranspiration over China” which seems to be very interesting. It considers daily scale PET in China across different zones and its contributing factors. It also gives ideas about trends in historical periods across the region. However, the authors used previous methods used by other people. Therefore, it is required to give more justification for certain results produced in a research paper. Furthermore, grammatical proofreading should be done by some experts to increase the drafting quality of the paper. I recommend minor to major comments for authors to improve the quality of research. I have given the comments in the Adobe Reader file. However, the authors are suggested to look into the following studies conducted on regional PET calculations and their related themes. These can give more ideas to authors to further clarify their results and interpretation of results more effectively.

1. https://doi.org/10.1007/s00382-023-06831-6

2. Spatiotemporal Analysis of Groundwater Resources Sustainability in South Asia and China using GLDAS Datasets." (2022).

6. PLOS authors have the option to publish the peer review history of their article (what does this mean?). If published, this will include your full peer review and any attached files.

Reviewer #1: No

Reviewer #2: No

---

## [Author Response · Author response to Decision Letter 0]

18 Dec 2023

Dear Dr. Hamed, 

Thank you very much for your letter dated on 22 November 2022 corresponding to our manuscript (PONE-D-23-30478) submitted to PLOS ONE.

We appreciate the two Reviewers' thoughtful comments and constructive suggestions, which are very helpful for improving the quality of our manuscript. We have gone through careful revision by addressing all the general and specific remarks. Please see the attached files that describe our changes and discussions. We are willing to receive the reviewers’ valuable comments once again if there is any.

Sincerely yours,

Wei

---

## [Decision Letter · Decision Letter 1]

12 Feb 2024

Revisiting the contribution of different factors in determining the changes in Potential evapotranspiration over China

PONE-D-23-30478R1

Dear Dr. Li,

We’re pleased to inform you that your manuscript has been judged scientifically suitable for publication and will be formally accepted for publication once it meets all outstanding technical requirements.

Kind regards,

Shamsuddin Shahid

Academic Editor

PLOS ONE

Additional Editor Comments (optional):

Reviewers' comments:

Reviewer's Responses to Questions

**Comments to the Author**

1. If the authors have adequately addressed your comments raised in a previous round of review and you feel that this manuscript is now acceptable for publication, you may indicate that here to bypass the “Comments to the Author” section, enter your conflict of interest statement in the “Confidential to Editor” section, and submit your "Accept" recommendation.

Reviewer #1: All comments have been addressed

Reviewer #2: All comments have been addressed

2. Is the manuscript technically sound, and do the data support the conclusions?

Reviewer #1: Yes

Reviewer #2: Yes

3. Has the statistical analysis been performed appropriately and rigorously? 

Reviewer #1: Yes

Reviewer #2: Yes

4. Have the authors made all data underlying the findings in their manuscript fully available?

Reviewer #1: Yes

Reviewer #2: Yes

5. Is the manuscript presented in an intelligible fashion and written in standard English?

Reviewer #1: Yes

Reviewer #2: Yes

6. Review Comments to the Author

Reviewer #1: The authors diligently addressed all the comments that were raised, demonstrating their commitment to incorporating constructive feedback and enhancing the quality of their work. Their thorough response reflects a conscientious effort to consider and incorporate suggestions, contributing to the refinement and overall robustness of their research or manuscript.

Reviewer #2: Thanks for addressing all the comments. Authors are appreciated for improving the quality of their publications. The manuscript may be accepted for publication.

7. PLOS authors have the option to publish the peer review history of their article (what does this mean?). If published, this will include your full peer review and any attached files.

Reviewer #1: No

Reviewer #2: No

---

## [Editor Report · Acceptance letter]

15 Feb 2024

PONE-D-23-30478R1 

PLOS ONE

Dear Dr. Li, 

I'm pleased to inform you that your manuscript has been deemed suitable for publication in PLOS ONE. Congratulations! Your manuscript is now being handed over to our production team.

Kind regards, 

on behalf of

Dr. Shamsuddin Shahid 

Academic Editor

PLOS ONE